# Serra da Estrela PDO Cheese Microbiome as Revealed by Next Generation Sequencing

**DOI:** 10.3390/microorganisms9102007

**Published:** 2021-09-22

**Authors:** Rui Rocha, Manuela Vaz Velho, Joana Santos, Paulo Fernandes

**Affiliations:** CISAS—Centre for Research and Development in Agrifood Systems and Sustainability, Escola Superior de Tecnologia e Gestão, Instituto Politécnico de Viana do Castelo, Rua Escola Industrial e Comercial de Nun’Álvares, 4900-347 Viana do Castelo, Portugal; mvazvelho@estg.ipvc.pt (M.V.V.); joana@estg.ipvc.pt (J.S.); paulof@estg.ipvc.pt (P.F.)

**Keywords:** traditional cheeses, Serra da Estrela cheese, protected designation of origin foods, next generation sequencing, microbiome, raw ewes’ milk, cardoon (*Cynara cardunculus* L.)

## Abstract

Serra da Estrela PDO cheese is the oldest traditional cheese manufactured in Portugal. In this work, its microbiome as well as the main raw materials used in cheese production, raw ewes’ milk and thistle flowers (*Cynara cardunculus* L.), were characterized using next generation sequencing. Samples were accordingly retrieved from a local producer over two consecutive production campaigns and at different time periods within each campaign. The bacterial and fungi communities associated with each matrix were accessed through sequencing of V3–V4 and Internal Transcribed Spacer 2 regions of rRNA gene amplicons, respectively. A high microbial diversity was found associated to each matrix, differing significantly (*p* < 0.05) from each other. Over 500 taxa were identified in each analyzed matrix, ranging from dominant (relative abundance > 1%), sub-dominant (0.01–1%) and rare taxa (<0.01%). Specifically, in cheese, 30 taxa were present in all analyzed samples (core taxa), including species of *Leuconostoc* spp. and *Lactococcus* spp. for bacteria and *Candida* spp., *Debaryomyces* spp. and *Yarrowia* spp. for fungi, that were cumulatively the most prevalent genera in Serra da Estrela PDO cheese (average relative abundance ≥10%). Ultimately, this characterization study may contribute to a better understanding of the microbial dynamics of this traditional PDO product, namely the influence of raw materials on cheese microbiome, and could assist producers interested in preserving the identity, quality and safety of Serra da Estrela PDO cheese.

## 1. Introduction

Serra da Estrela cheese (SEC) is the most renowned Portuguese traditional cheese [1]. Produced in the foothills of the Serra da Estrela mountain, to which it owes its name, it follows artisanal protocols passed down between generations of shepherds, tracing back to the Roman occupation of the Iberian Peninsula [1,2,3]. Nowadays, SEC is the primary source of income of local shepherds and farmers, and is one of the most important economic activities in the region [1,2,3].

SEC manufacturing uses ovine raw milk from pasture feed females of “Bordaleira Serra da Estrela” and/or “Churra Mondegueira” breeds, coagulated with the addition of crude extracts of dried *Cynara cardunculus* L. flower and salt [4]. After a ripening period of 30–45 days, a flat cylinder, highly aromatic with a clean, smooth and slightly acidically flavored cheese is obtained. Featuring a thin, soft, uniform straw–yellowish rind and an ivory–white, buttery, creamy and unctuous paste with none too few little eyes [1,2,3,4]. These intrinsic characteristics earned it, in 1996, the Protected Designation of Origin (PDO) status.

Over the years, extensive fundamental and applied research has been carried out to unveil SEC unique organoleptic characteristics (reviewed in Inácio et al. [4] and Macedo et al. [2]). One critical factor is the use of raw untreated milk over the deliberate addition of starter cultures. This means that the native microbiota of the milk ends up playing a critical role in the manufacture of SEC [2,4,5]. The microbial consortia associated with SEC has been extensively studied, from the raw materials up to the final product [4]. However, these studies were limited to data gathered from cultured dependent methods [3,5,6,7,8], which are unable to truly unveil the diversity associated with these types of product [9,10,11]. Over the last decade, culture-independent methods, such as high-throughput sequencing technologies, have emerged, allowing an in-depth characterization and the identification of both high and low abundance microbiota at a species and, even in some cases, at a subspecies level [12,13,14,15,16]. Hence, Next Generation Sequencing (NGS) is been applied in the characterization of the microbial communities of traditional cheeses [9,17,18,19,20,21]. However, as far as the authors know, this is the first time that is used to characterize SEC microbiome. Therefore, the aim of the present work was to characterize the microbial community, of both bacteria and fungi, associated with the SEC and raw materials using NGS.

## 2. Materials and Methods

### 2.1. Cheese Manufacture and Sampling Strategy

For microbial characterization of SEC, raw ewes’ milk (REM), cardoon and cheese samples were analyzed from two consecutive production campaigns (1) 2018/2019; (2) 2019/2020. These were subdivided into three periods: (1) November–January; (2) February–March; and (3) May–June, as depicted in Figure 1.

The sampling strategy allowed a systematic analysis of SEC production, including autumn, winter and spring manufactures, reflecting distinct weather and pasture conditions that, on previous reports, posed a significant influence on microbial abundance [2,6,7]. Furthermore, to account for farmhouse variability REM from the two producers were used and analyzed throughout the study, A in 2018/2019 and B in 2019/2020 [2]. Moreover, four types of cardoon from *Cynara cardunculus* L. were used and analyzed. One was provided by the cheese producer and is regularly used in SEC manufacture, identified as “commercial”, and three genotypes of *Cynara cardunculus* L.,1 M, 3 M and 6 M [22].

Finally, SEC was manufactured in a certified PDO cheese producer from Gouveia, Portugal. REM and cardoon samples used for cheese manufacture were collected immediately before use. Serra da Estrela cheeses were produced following standard procedures and a ripening period of 38 days. All samples were transported and kept under refrigeration (≤4 °C) before processing.

### 2.2. Total DNA Extraction

Upon arrival, milk, cardoon and cheese samples were prepared for DNA extraction. For REM, three replicate tubes containing each 6 mL of milk were pelleted at 13,000 rpm (Hettich, Tuttlingen, Germany) for 10 min [23]. The supernatant was discarded and the pellet was resuspended in DNA/RNA Shield™ solution (ZymoResearch, Irvine, CA, USA). For cardoon, 2 to 10 g of dried thistle flower samples were suspended in 100 mL of a sterile solution containing 10% (*v/v*) Buffered Peptone Water (Liofilchem srl, Roseto degli Abruzzi, Italy) and 0.01% (*v/v*) Tween 80^®^ (PanReac AppliChem). Cardoon suspension was subjected to an ultrasonic bath for 30 s (Jet Program option) (Soltec, Milan, Italy) followed by 150 rpm orbital agitation for 30 min at room temperature (B. Braun Biotech International, Melsungen, Germany). Plant material was grossly filtered out using a stomacher bag filter and the remaining solution was divided into three 20 mL replicate aliquots. Cells were pelleted at 10,000 rpm for 5 min, the supernatant was discarded, and the pellet was resuspended in a DNA/RNA Shield™ solution. For cheese, three replicate portions of 400 mg were aseptically retrieved from the paste side and center of each cheese and were transferred to a tube containing a DNA/RNA Shield™ solution. All sample replicates of milk, cardoon and cheese in DNA/RNA Shield™ solution were homogenized in a bead beater (Benchmark Scientific, Sayreville, USA) for 6 min at maximum speed and stored at −20 °C until DNA extraction.

Total DNA was extracted from each sample replicate using ZymoBIOMICS™ DNA Miniprep Kit (ZymoResearch, Irvine, CA, USA), following the manufacturer’s instructions. The eluted DNAs were cleaned and concentrated using a DNA Clean and Concentrator™ Kit (ZymoResearch, Irvine, CA, USA), following the manufacturer’s instructions. Finally, the purified DNA from three independent extractions of each sample was pooled together and used in downstream procedures.

### 2.3. Sequencing Preparation, Run and Processing

The pooled DNA from each sample was subjected to two separated PCR amplification runs, one for the V3–V4 hypervariable region of the 16S rRNA gene and the other for the Internal Transcribed Spacer (ITS) region of the 18S rRNA. The PCR amplification runs were performed using KAPA HiFi HotStart PCR Kit (Roche, Indianapolis, IN, USA) according to manufacturer instructions, using 0.3 µM of each primer and 12.5 ng of template DNA, in a total volume of 25 µL. For the amplification of the V3–V4 region, the forward Bakt_341F 5′–CTACGGGNGGCWGCAG–3′ and reverse primers Bakt_805R 5′–GACTACHVGGGTATCTAATCC–3′ were used [24,25]. For the amplification of the ITS region, a pool of forward primers, ITS3NGS1_F 5′–CATCGATGAAGAACGCAG–3′, ITS3NGS2_F 5′–CAACGATGAAGAACGCAG–3′, ITS3NGS3_F 5′–CACCGATGAAGAACGCAG–3′, ITS3NGS4_F 5′–CATCGATGAAGAACGTAG–3′, ITS3NGS5_F 5′–CATCGATGAAGAACGTGG–3′, ITS3NGS10_F 5′–CATCGATGAAGAACGCTG–3′ and reverse primer ITS4NGS001_R 5′–TCCTSCGCTTATTGATATGC–3′ were used [26]. PCR conditions included a 3 min denaturation at 95 °C, followed by 25 cycles of 98 °C for 20 s, 55 °C (V3–V4 region)/60 °C (ITS region) for 30 s and 72 °C for 30 s and a final extension at 72 °C for 5 min. Then, a second PCR reaction added indexes and sequencing adapters to both ends of the amplified target region according to the manufacturer’s recommendations [27]. Negative PCR controls were included for all amplification procedures. PCR products were finally purified and normalized using SequalPrep 96-well plate kit (ThermoFisher Scientific, Waltham, MA, USA) [28], pooled and pair-end sequenced in the Illumina MiSeq^®^ sequencer with the V3 chemistry, according to the manufacturer’s instructions (Illumina, San Diego, CA, USA) at Genoinseq (Cantanhede, Portugal).

Raw reads were extracted from Illumina MiSeq^®^ System in FASTQ format and quality-filtered with PRINSEQ version 0.20.4 [29] to remove sequencing adapters, trim bases with an average quality lower than Q25 in a window of 5 bases and reads with less than 100 and 150 bases in length for ITS and V3–V4 files, respectively. Forward and reverse reads were merged by overlapping paired end reads with AdapterRemoval version 2.3.0 [30] using default parameters. ITSx version 1.1.2 [31] was used on ITS files to extract the highly variable fungal ITS2 subregion from merged reads and were then filtered to remove ITS2 reads below 100 bases in length. Sample IDs were assigned to the merged reads and were converted to a FASTA format. Chimeric merged reads were detected and removed using VSearch [32], an implementation of UCHIME [33] against Greengenes database version 13_8 [34] for V3–V4 and UNITE/QIIME ITS database version 8.2 [35] for ITS2 files. Operational Taxonomic Units (OTU) were generated in the Quantitative Insights into Microbial Ecology (QIIME) software version 2020.2 [36]. OTUs were selected at a similarity cut-off of 97% using the open reference strategy and those with less than two reads were discarded.

### 2.4. Bioinformatics and Statistical Analysis

All files generated in the previous section were analyzed in QIIME software version 2020.8 [36]. Diversity indexes namely, Shannon [37], Simpson [38] and Goods Coverage [39], as well as richness estimators, Chao1 [40] and Abundance Coverage Estimator (ACE) [41], were determined using *q2–diversity* plugin and *alpha* pipeline.

A phylogenetic tree was generated using *q2–phylogeny* plugin and *align–to–tree–mafft–fasttree* pipeline [42,43]. From there, OTU, Faith Phylogenetic diversity [44] and Shannon [37] rarefaction plots were constructed using *q2–diversity* plugin and *alpha–rarefaction* pipeline (Appendix A). Furthermore, *q2–diversity* plugin and *core–metrics* pipeline was used to: (1) calculate α–diversity metrics, namely Shannon, observed features, Faith phylogenetic diversity and Evenness; (2) calculate β–diversity metrics, namely Jaccard distance [45], Bray–Curtis dissimilarity [46] and unweighted and weighted UniFrac [47,48]; (3) generate principal coordinates analysis (PCoA) plots. The sampling depth parameters used in this step were defined for each group of samples to be the highest possible, while retaining all samples (Appendix A). However, due to a very low number of reads (8832), the 6M cardoon sample from the 2019/2020 campaign was intentionally dropped from the ITS diversity analysis between matrices. The generated files were further analyzed within *q2–diversity* plugin and *alpha–group–significance* and *beta–group–significance* pipelines to explore and disclose α and β–diversity differences, using Kruskal–Wallis [49] and Permutational multivariate analysis of variance tests [50], respectively, between matrices, campaigns, periods and cardoon genotypes. Differences with *p* < 0.05 were considered significant.

OTU taxonomy assignment was accomplished using *q2–feature–classifier* plugin [51]. A series of trial classification runs were made to evaluate the best taxonomy assignment in sequence files obtained from a mock community sample (ZymoResearch, Irvine, CA, USA) sequenced in parallel with the samples. To that end, *classify–sklearn* and *classify–consensus–vsearch* pipelines were tested using GreenGenes (version 13_8) [34], NCBI (BioProject 33175) (U.S. National Library of Medicine, Bethesda, USA), EZBioCloud (version of January_2021) (ChunLab, Seoul, Republic of Korea), ARB Silva (version SSURef_NR99, release 138) [52] and UNITE (version 8.2, dynamic fungi release) [35] databases. The best taxonomic outcome for V3–V4 mock file was obtained using the *classify–consensus–vsearch* pipeline coupled with ARB Silva database and a confidence threshold for taxonomic assignment ≥90%. For mock ITS files, the best performance was obtained using *classify**–sklearn* pipeline coupled with a UNITE database and a confidence threshold for taxonomic assignment ≥70%. These taxonomic annotations were used throughout the work for the construction of stacked bar charts, identification of core taxa by matrix and to identify the taxa responsible for α and β–diversity differences. A Linear Discriminant Analysis (LDA) of Effect Size (LEfSe) was used to estimate the effect size on differentially abundant taxa between and within matrices [53]. LEfSe was performed on a Galaxy computational tool (http://huttenhower.sph.harvard.edu/galaxy/, accessed on 7th of April 2021) using a 0.05 alpha value for the factorial Kruskal–Wallis test among classes, a 0.05 alpha value for the pairwise Wilcoxon test between subclasses, a “one against all” strategy for multi-class analysis, a pairwise comparison only among subclasses with the same name and a logarithmic LDA score threshold of 3.5 and 6.0 for bacteria and fungi, respectively. Observed abundances, at any given taxonomic level, of the total reads were then used to define the dominant (>1%), sub-dominant (between 0.01 and 1%) and rare taxa (<0.01%) [16].

### 2.5. Nucleotide Sequences Accession Number

Raw reads were deposited in SRA database under BioProject PRJNA723623.

## 3. Results and Discussion

### 3.1. Microbial Community of Raw Ewes’ Milk

Milk present in the upper part of a healthy lactation female is generally considered sterile [54,55]. Milk colonization occurs from direct and/or indirect transfer from reservoirs. Direct reservoirs are locations and/or surfaces that at some point contact directly with the previous sterile raw milk, such as the teat canal, teat surface and milking equipment. Indirect reservoirs can be, for example, feed material, grass, soil, air, litter, water (drinking and/or washing), stable, milker and milk parlour [15,54,55].

In this study, over 500 genera (213 of bacteria and 293 of fungi) were identified in the microbial community of REM (Appendix A). This composition is in general, analogous to previous reports on the microbiology of REM from Serra da Estrela region [5], from other locations [56,57,58,59,60,61] and raw milk in general [15,54,55,62].

Despite a wide taxa variability between milk samples, α and β group significance analysis showed that none of the possible factors, such lactation period/season and producer/campaign, posed a significant impact on sample diversity (*p* > 0.05). Except for the number of fungi OTUs, which were significantly higher in milk retrieved from producer A (2018/2019 campaigns) in comparison to the one from producer B (2019/2020 campaigns) (*p* < 0.05).

The genus *Candida*, frequently related to yeast udder inflammation [63,64], with 56% of the relative frequency on average, and *C. zeylanoides* with 40% were the predominant fungi taxa in REM (Figure 2 and Appendix A), despite the procedures for the production of certified SEC banning the use of ovine milk obtained from unhealthy females [65]. This high prevalence, without any other information, could be indicative of latent infection, which corresponds to 95–98% of all mastitis cases [63]. Hence, a constant monitorization of the health condition of lactating females is advised to prevent economic losses throughout the production chain.

Psychrotolerant microflora accounted for ≈40% of total bacterial counts (Figure 2 and Appendix A). This is most likely the result of a combination of milk refrigeration, as recommended in the manufacture manual of SEC, for transportation and storage purposes and milk handling in cold weather, typical from the region of Serra da Estrela [65]. These conditions are known to alter the balance of milk microflora, from a gram-positive to a gram-negative dominance [15,54,55,66]. Finally, Lactic Acid Bacteria (LAB) accounts for a quarter of bacterial counts, with *Lactococcus lactis*, *Enterococcus faecium* and *Leuconostoc mesenteroides* with 12, 4 and 3% of the relative frequency on average, respectively, as the most prevalent species (Figure 2 and Appendix A).

### 3.2. Microbial Community of Cardoon

The dried thistle flower of *Cynara cardunculus* L., cardoon is frequently used as a clotting agent in the production of goat and ovine milk cheeses in Portugal, Spain and Italy [67]. The addition of cardoon in cheese production is not standardized, but is still an empirical exercise of the cheese maker, ranging from 0.15 to 0.6 g of flower in L^−1^ of milk [4,67]. Furthermore, several methods are reported for the addition of cardoon in cheese making [2]. One of the most popular, and the one used in the selected cheese factory, consists of the maceration of cardoon flowers in water, filtering this extract with the help of a fine clean cloth, which is poured directly into the warm milk. Extensive research was carried out to study the influence of cardoon in cheese production, from a chemical, biochemical, microbiological and sensorial point of view (reviewed in Conceição et al. [67]). However, only a few reports accessed the microbial composition of cardoon extracts [68,69,70].

Over 500 genera (265 of bacteria and 263 of fungi) were identified in cardoon samples, comprised mostly by taxa with a ubiquitous distribution within the environment and frequently isolated from vegetal material such *Aspergillus*, *Aureobasidium*, *Cladosporium*, *Lactobacillus*, *Mucor*, *Pantoea*, *Pseudomonas* and *Rhizopus* (Figure 3 and Appendix A) [55,71,72,73,74,75].

Despite a wide taxa variability between cardoon samples, α and β group significance analysis showed that none of the possible factors, such cardoon genotype and/or production campaign, posed a significant impact on sample diversity (*p* > 0.05). Except for the number of bacterial OTUs, which were significantly higher in cardoon samples from 2019/2020 campaign in comparison to the ones from 2018/2019 (*p* < 0.05).

### 3.3. Microbial Community of Serra da Estrela Cheese

The microbial community of cheese is composed of variable ranges of bacteria, yeasts and molds [54,62,76,77]. Their access to cheese occurs either by deliberate addition, transference from cheese ingredients (e.g., milk, rennet, salt, among others) and/or transference from direct contact with equipment and the cheese maker [76,78].

The microbiological characterization of SEC samples revealed the presence of almost 500 genera (138 of bacteria and 338 of fungi—Appendix A). A composition that is in general, analogous to previous SEC characterizations [3,5,6,7,8,79], other ovine milk cheeses [18,21,79,80,81,82,83,84], and cheese in general [54,62,77].

As expected, the bacterial community is mainly composed of LAB, ranging from 50 up to 90% of all counts (Figure 4 and Appendix A). Of those, *Leuconostoc mesenteroides* and *Lactococcus lactis* stand out as the major constituents. This is in contrast with previous characterizations, that have shown a limited contribution of these species, in particular to the overall microbial composition of SEC [5,6]. However, recent publications concerning other ovine milk cheeses show a high prevalence of *L. mesenteroides* and *L. lactis*, namely Azeitão from Portugal [80], Krčki, Istrian and Paski from Croatia [18], Feta from Greece [84] and Fiore Sardo from Italy [82].

The fungi community was dominated by the presence of *Candida sake*, *Candida zeylanoides* and *Debaryomyces hansenii* yeasts (Figure 5 and Appendix A). *D. hansenii* is, in fact, one of the most prevalent yeasts found in cheese, previously described as a major constituent of the SEC yeast community [5,6] and other ovine milk cheeses [21,62,77,83,85]. On the contrary, the appearance and role of *Candida* spp. in cheese is a controversial topic. Generally regarded as a cheese contaminant, some species are known to be opportunistic pathogens (e.g., *C. parapsilosis*, *C. tropicalis*, *C. albicans*, among others), while others are nowadays used as adjunct cultures (e.g., *C. krusei* and *C. colliculosa*) [62]. *Candida* spp. have been previously identified as part of SEC microbiota, namely *C. rugosa*, *C. zeylanoides* and *C. etchellsii* [5]. Moreover, recent microbial characterization studies have highlighted the widespread presence of *Candida* species in cheese, which has led some authors to raise a yet-undiscovered role during cheese ripening stage [21,62,81,86,87]. In this case, its high prevalence results, most likely, from the high prevalence observed in ovine milk samples, as previously detailed.

Concerning food safety, the analyzed SEC samples were mostly colonized by beneficial and probiotic microbiota. Although, foodborne pathogens, responsible for occasional foodborne disease outbreaks derived from the consumption of contaminated cheese products, have also been detected (reviewed in Fox et al. [88]). Pathogenic bacteria such as *Clostridium perfringens*, *Listeria monocytogenes*, *Yersinia enterocolitica*, *Salmonella enterica* and *Staphylococcus aureus* were found, but at rare (<0.01%) or near the rare range (<0.09%) (Appendix A), attesting the overall safety of SEC.

Interestingly, an analysis of the abundance histograms points towards a higher diversity of 2019/2020 vs. 2018/2019 cheeses, while some taxa appear to be season- and campaign-specific (Figure 4 and Figure 5). In fact, α and β group significance analysis showed that the production campaign (2018/2019 vs. 2019/2020), as well as the manufacture period (November–January vs. February–March vs. May–June), poses a significant impact on SEC sample diversity (*p* < 0.05). Similarly, previous studies have shown the influence of the dairy farm, production campaign, season and axial location factors on overall microbial composition of SEC [2,3,5,6,7]. An LEfSe analysis on the data revealed 31 divergent taxa between campaigns (Figure 6). Of those, most of the divergent taxa were more associated to autumn manufactures, except for *Kocuria* spp., *Lactococcus* spp., *Carnobacterium* spp., *C. curvatus* and *K. servazii*, which were more abundant in winter and *L. lactis*, *Lactobacillus* spp., *Acinetobacter* spp., *Citrobacter* spp., *S. stellimalicola* and *C. metapsilosis* in spring cheeses (Appendix A). Overall, these differences are likely to reflect a higher/lower presence of each specific taxon as a resident microbiota in the REM used for each cheese manufacture (Appendix A).

### 3.4. Matrices Taxa Variability and Core Community

The microbial composition and diversity associated with cheese, curd, whey, milk and the other used ingredients in cheese manufacturing, differ significantly between each other, reflecting each matrix type intrinsic characteristics and/or interactions [14]. Unsurprisingly, α and β group significance analysis showed that overall, REM, cardoon and SEC samples differ significantly in their microbial composition (*p* < 0.05). α group significance analysis showed that cheese samples present more homogenous microbial communities, while cardoon samples are the most diverse matrix. Finally, milk samples presented a high bacterial and low fungi diversity, similar to those observed in cardoon and cheese samples, respectively. Moreover, β group significance analysis showed that is possible to discriminate between each matrix, based on their specific bacterial community (*p* < 0.001), whereas the fungi community can only be used to distinguish cardoon from the grouped milk and cheese samples (*p* < 0.001) (Appendix A). Overall, through an LEfSe analysis, it was found that psychrophiles, LAB and yeasts and plant-associated microorganisms are the most discriminatory biomarkers for REM, SEC and cardoon, respectively (Figure 7 and Figure 8).

The abundance data allowed the determination of the core microbial community associated with each matrix (Figure 9 and Figure 10), that ranged from dominant, sub-dominant to rare taxa (Appendix A). As demonstrated by α diversity metrics, cardoon is by far the matrix with the most diverse core microbial community with 61 taxa, followed by REM with 47 and SEC with 30. As expected, SEC shared a higher number of core taxa with REM than with cardoon—18 vs. 11—and only 9 were shared between all matrices. *L. lactis*, one of the most prevalent species in SEC, is one of the most important LAB species in the dairy industry [80]. Its presence in SEC derives predominantly from REM (accounting up to 12% of total counts) and rarely from cardoon (≤0.5% of total counts). It belongs to a restricted group of microorganisms, denominated starter bacteria, whose main function is the production of lactic acid from lactose metabolization [89]. Furthermore, it contributes to the development of the organoleptic characteristics of cheese during ripening, with the production of flavor compounds (e.g., free amino acids and medium- and small-peptides) from the metabolization of milk proteins and to texture through the secretion of exopolysaccharides [90,91].

Likewise, *L. mesenteroides* composes a significant portion of the SEC microbial community. In contrast to the low abundance levels observed in the raw materials of origin, REM and cardoon, respectively with 4 and 1% of total counts on average, resulting from a poor growth in milk [55]. To stimulate growth, leuconostocs requires the addition and/or synthesis of peptides and amino acids by other microorganisms and, as result of that, are frequently found in close symbiosis with lactococci in the microbiota of fresh and semi-hard cheeses [55,92]. Active during ripening, *L. mesenteroides* are involved in the development of cheese organoleptic properties through the production of aromatic compounds from co-metabolization of lactose and citrate [55,78].

Enterococci, namely *E. faecalis* and *E. faecium*, are another group of core microbiota in SEC. However, they are present at relatively low abundances when compared to other LABs (overall ≤5% of the relative frequency on average). These are a controversial group within LAB, that, depending on the strain, can be considered as a starter culture, probiotic, spoilage, or pathogenic bacteria [93]. They can grow under different conditions and substrates and survive refrigeration, high-temperature and high-salinity environments. However, SEC seems to provide an unfavorable growth environment, either by microbial competition and/or substrate deficiency, since its load on cheese is similar to the ones observed in the raw materials of origin, REM and cardoon (≈4 vs. ≤1%, respectively). Furthermore, the dominance of *E. faecium* over other enterococci species is maintained across matrices. Nonetheless, enterococci are known to contribute to the development of the organoleptic characteristics of cheese through their proteolytic, lipolytic, and citrate breakdown activities [55,77,78].

*L. curvatus* and *L. plantarum* compose the lactobacilli core microbiota of SEC, present at a sub-dominant level (0.01 ≤ relative abundance ≤ 1%), with the exception of *L. curvatus* on 2019/2020 cheeses that were found with a relative frequency of ≈5% on average. These lactobacilli are only occasionally found in REM and cardoon, which supports an external source of these taxa. They are active during the cheese ripening stage and are involved in a series of events, namely residual sugar metabolization, peptide hydrolysis, amino acid conversion, production of flavor and aroma compounds and the production of antifungal and antibacterial compounds [77].

Salt tolerant bacteria such *Hafnia* spp., *Vibrio* spp. and micrococci were found to range between sub-dominant and dominant levels among SEC samples. These taxa are known to survive the brining process and actively contribute to the development of the organoleptic characteristics of cheese [15,78]. Micrococci, and to a lower extent *Hafnia* spp., were found to inhabit REM and cardoon microflora, while *Vibrio* spp. was practically absent from these raw materials. An observation that points towards an equipment/facility-related input, for example during cheese washing and salting steps [13].

*Psychrobacter* spp. was found at a near rare range within SEC, and only occasionally in REM and cardoon samples. Capable of producing aldehydes, ketones and sulphur-volatile compounds, it contributes to the development of cheese organoleptic characteristics [55]. Overall, *Enterobacteriaceae* composes of a noteworthy group of REM, cardoon and SEC microflora. Despite their high aromatic potential, the presence of *Serratia* spp., *Enterobacter* spp. and *Citrobacter* spp. is often linked to detrimental effects, such hydrogen gas production and off-flavor development [15,54].

Finally, the fungi core taxa of SEC are mostly composed by yeasts (Figure 10). These are important during cheese ripening, namely in matrix deacidification through lactate consumption and in the development of cheese flavor and aroma. Yeast colonization of cheese occurs predominantly in the rind or in its vicinity, apart from some fermenting species that can grow in the cheese interior [13,62,77]. *D. hansenii* is a natural inhabitant of cheese, owing to its ability to metabolize lactate, lactose and tolerance to acidic and hypersaline habitats [62,77]. As previously stated, it is the third most prevalent yeast in SEC, whose strains arise predominantly from REM (≈6% of total counts) and to a smaller extent from cardoon (≤1%). For *Y. porcina*, *K. lactis* and *S. cerevisiae*, however, REM acts most likely as the primary source of strains for SEC, as it is virtually absent in cardoon samples. *K. lactis* plays an important role at the beginning of cheese ripening, namely in lactose metabolization, ethanol production and in the secretion of lytic enzymes, hereby contributing to cheese flavor and aroma [15,55,77]. *S. cerevisiae*, occasionally found in cheese, is active during the ripening stage, when it metabolizes lipids and proteins, contributing to cheese flavor and aroma [77]. On the contrary, *Y. porcina* is not commonly isolated from cheese, unlike *Y. lipolytica* [62,77,94]. *Y. lipolytica* presents a strong lipolytic and proteolytic activity, thus contributing to the organoleptic characteristics of cheese. Accordingly, it is possible that *Y. porcina*, similarly to the role of *Y. lipolytica* in other cheeses, is involved in flavor and aroma development in SEC.

## 4. Conclusions

To the extent of our knowledge, this study represents the first application of next generation sequencing technology to the characterization of the microbial community associated with the Serra da Estrela PDO cheese. This comprehensive work provided an insight into raw materials, raw ewes’ milk and cardoon, used in cheesemaking and cheeses analyzed over two consecutive production campaigns and three periods within each campaign. The here-disclosed microbial profiles, not just of the cheese but also of the milk and thistle flowers used in cheese manufacture, display a high microbial variability and have offered some insights regarding the provenience and abundance patterns of the microbiota found in cheese. Ovine milk was found as the most probable strain source for the core taxa revealed in SEC, with limited contributions of cardoon. Lactobacilli and *Vibrio* spp. were virtually absent from the raw materials analyzed, suggesting the contribution of other sources to the overall SEC microbiome. Moreover, the presence a newly described yeast, *Y. porcina*, not commonly associated with cheese, is revealed. This observation requires further investigation, namely in the evaluation of its impact on cheese microbiota dynamics and its contribution to cheese organoleptic characteristics.

Overall, this study contributes to a better understanding of a complex, valuable and appreciated product that must meet rigorous criteria for authenticity and PDO certification. Nonetheless, future research could focus on the expansion of the scope of this characterization, assessing microbial axial location, dairy-farm to dairy-farm variability, and microbial dynamics during cheese production and ripening stages, potentially addressing some of the unanswered questions raised in this work.

## Figures and Tables

**Figure 1 microorganisms-09-02007-f001:**
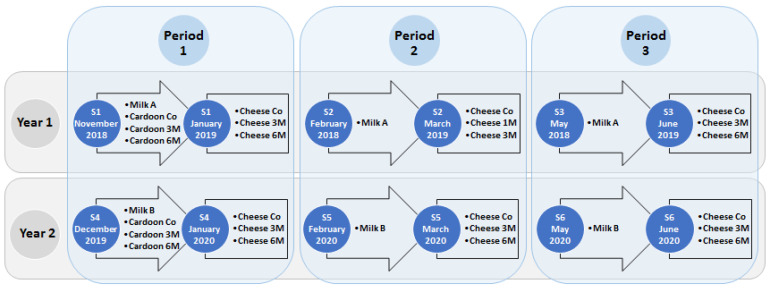
Schematic representation of the sampling strategy used for the microbial characterization of ewes’ raw milk, cardoon and Serra da Estrela PDO cheese. Co refers to the commercially used cardoon by the cheese producer and 1 M, 3 M and 6 M refers to specific *Cynara cardunculus* L. genotypes used in this study in the manufacture of Serra da Estrela cheese.

**Figure 2 microorganisms-09-02007-f002:**
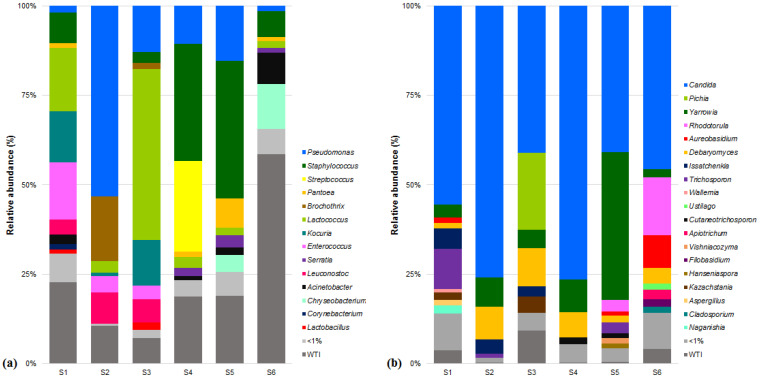
Relative abundances (%) of dominant sequences (>1%) assigned to genus level identified in raw ewes’ milk DNA samples based on partial sequence analysis of the (**a**) V3–V4 and (**b**) Internal Transcribed Spacer 2 regions of the rRNA gene. WTI refers to percentage of sequences without taxonomic attribution to the specified taxonomic level.

**Figure 3 microorganisms-09-02007-f003:**
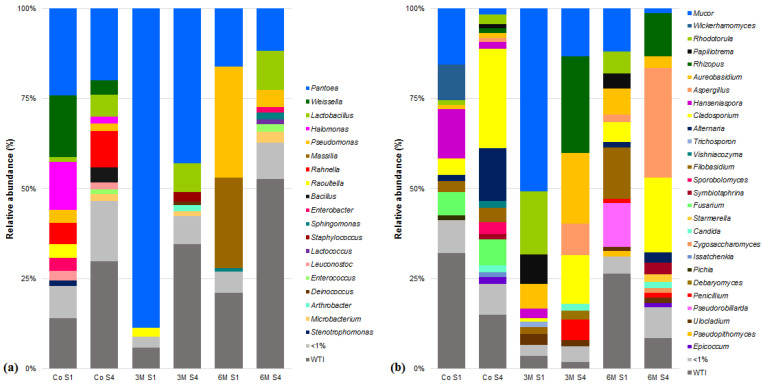
Relative abundances (%) of dominant sequences (>1%) assigned to genus level identified in dried flowers of *Cynara cardunculus* L. (cardoon) DNA samples based on partial sequence analysis of the (**a**) V3–V4 and (**b**) Internal Transcribed Spacer 2 regions of the rRNA gene. Co refers to the commercially used cardoon by the cheese producer and 3 M and 6 M refers to specific cardoon genotypes. WTI refers to the percentage of sequences without taxonomic attribution to the specified taxonomic level.

**Figure 4 microorganisms-09-02007-f004:**
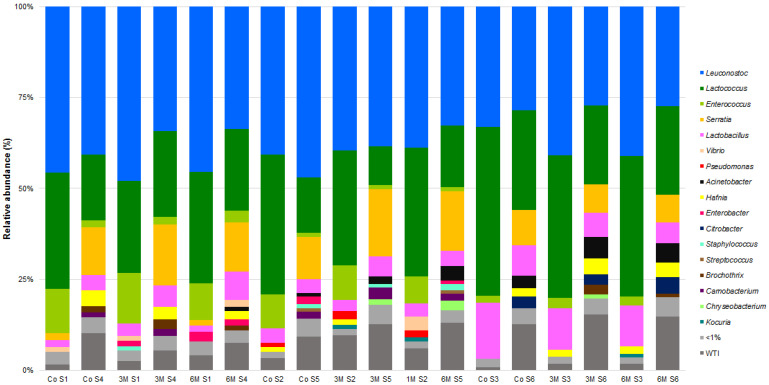
Relative abundances (%) of dominant sequences (>1%) assigned to genus level identified in Serra da Estrela cheese DNA samples based on partial sequence analysis of the V3–V4 region in the rRNA gene. Co refers to the commercially used cardoon by the cheese producer and 1 M, 3 M and 6 M refers to specific cardoon genotypes. WTI refers to the percentage of sequences without taxonomic attribution to the specified taxonomic level.

**Figure 5 microorganisms-09-02007-f005:**
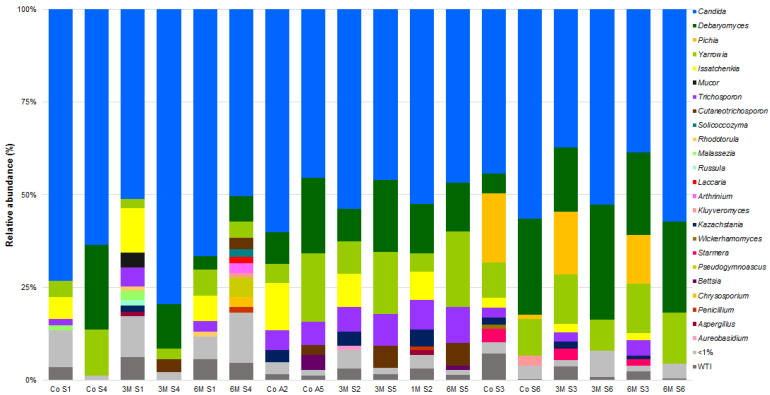
Relative abundances (%) of dominant sequences (>1%) assigned to genus level identified in Serra da Estrela cheese DNA samples based on partial sequence analysis of the Internal Transcribed Spacer 2 region in the rRNA gene. Co refers to the commercially used cardoon by the cheese producer and 1 M, 3 M and 6 M refers to specific cardoon genotypes. WTI refers to the percentage of sequences without taxonomic attribution to the specified taxonomic level.

**Figure 6 microorganisms-09-02007-f006:**
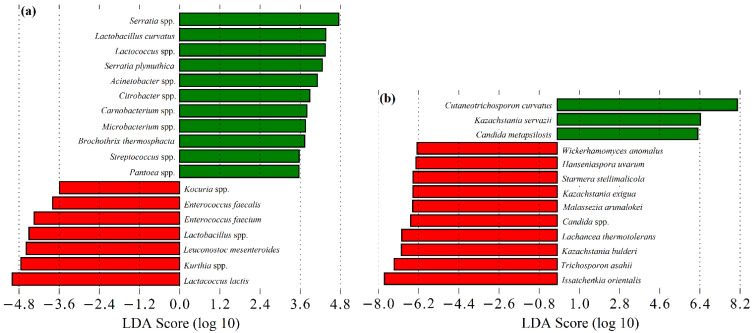
Linear Discriminant Analysis (LDA) scores computed for: (**a**) bacterial and (**b**) fungi features differentially abundant in Serra da Estrela cheese from: (**a**) 2018/2019 (red); (**b**) 2019/2020 (green) campaigns.

**Figure 7 microorganisms-09-02007-f007:**
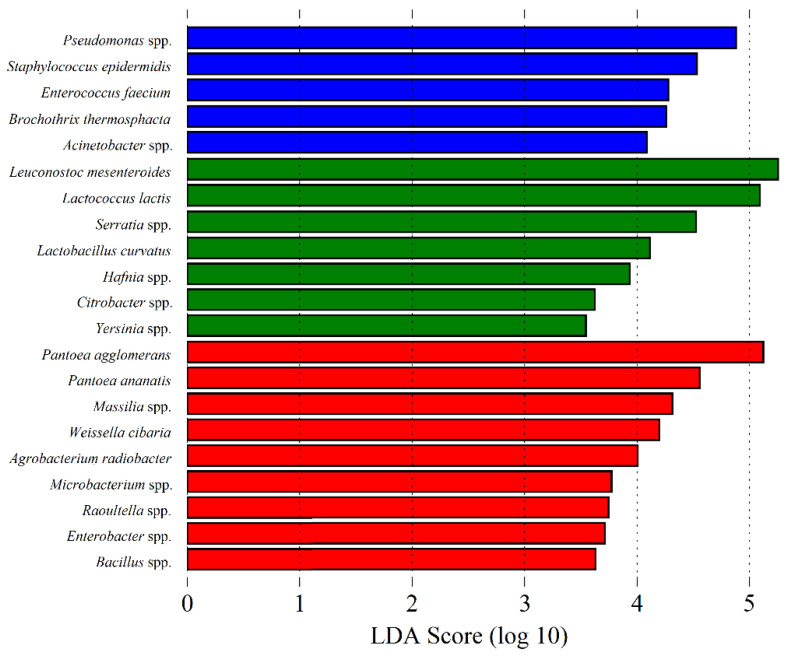
Linear Discriminant Analysis (LDA) scores computed for bacterial features differentially abundant in: raw ewes’ milk (blue bars); Serra da Estrela cheese (green bars); and dried flowers of *Cynara cardunculus* L. (cardoon (red bars)).

**Figure 8 microorganisms-09-02007-f008:**
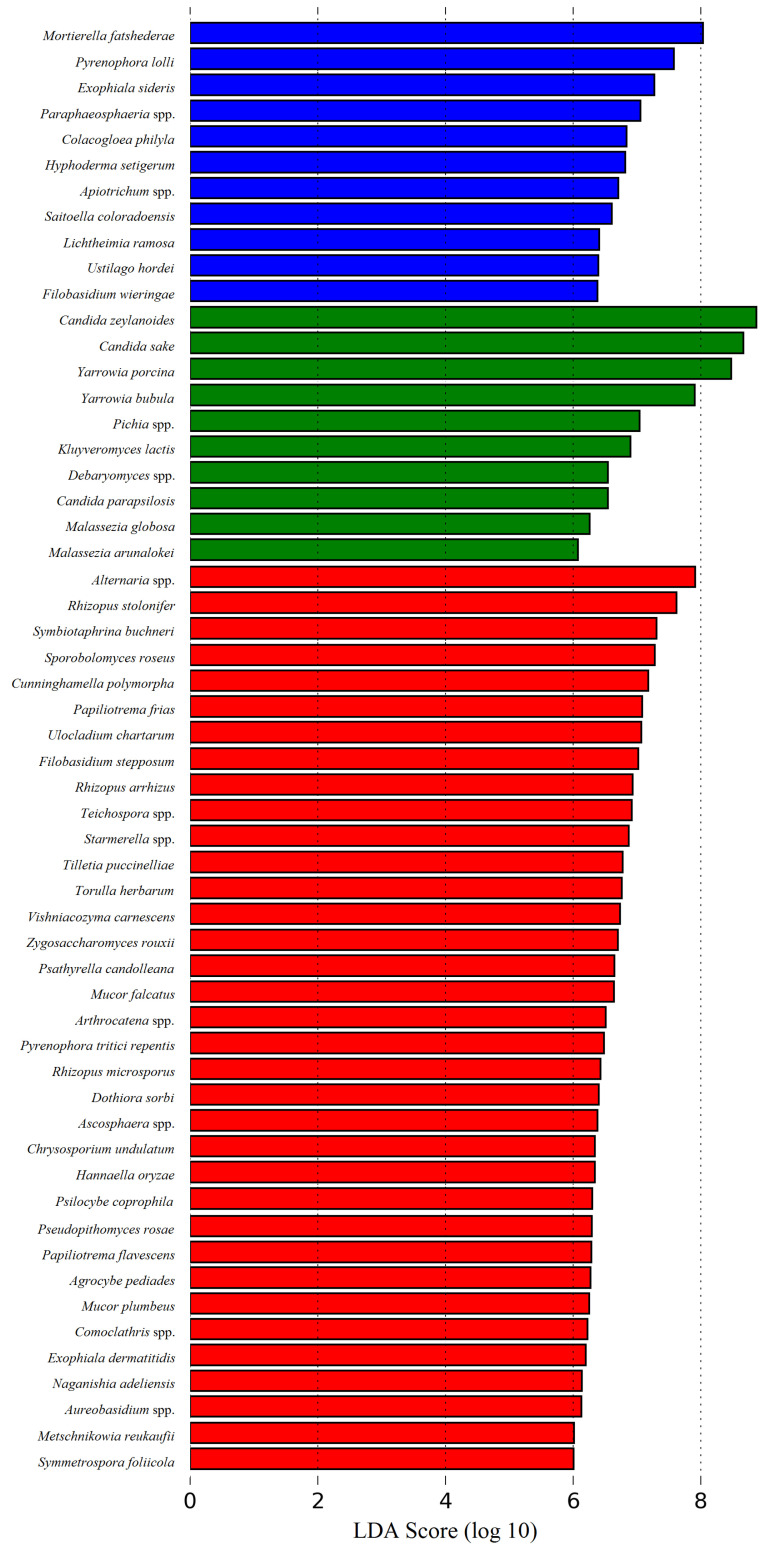
Linear Discriminant Analysis (LDA) scores computed for fungi features differentially abundant in: raw ewes’ milk (blue bars); Serra da Estrela cheese (green bars); and dried flowers of *Cynara cardunculus* L. (cardoon (red bars)).

**Figure 9 microorganisms-09-02007-f009:**
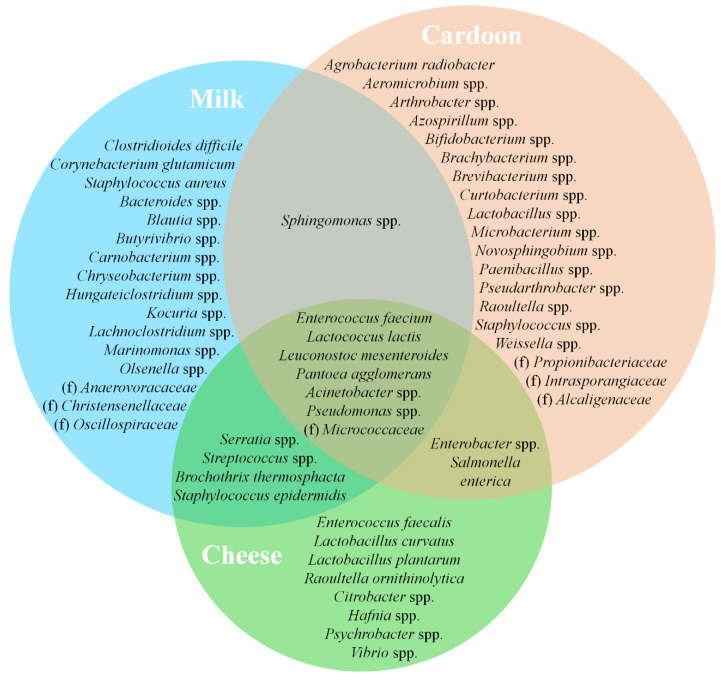
Venn diagram representing shared core taxa of bacteria between raw ewes’ milk (blue area), cardoon (red area) and Serra da Estrela cheese (green area) matrices. f denotes identified bacterial families without taxonomic attribution to genus and/or species.

**Figure 10 microorganisms-09-02007-f010:**
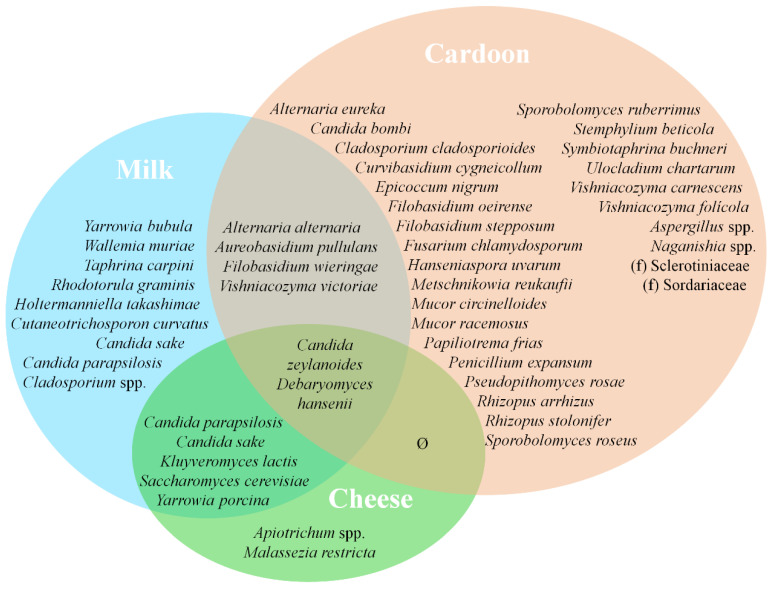
Venn diagram representing shared core taxa of fungi between raw ewes’ milk (blue area), cardoon (red area) and Serra da Estrela cheese (green area) matrices. f denotes identified bacterial families without taxonomic attribution to genus and/or species.

## Data Availability

Raw reads were deposited in SRA database under BioProject PRJNA723623.

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
