# Peer review of "Serra da Estrela PDO Cheese Microbiome as Revealed by Next Generation Sequencing"

_microorganisms, 2021, doi:10.3390/microorganisms9102007_

Round 1

Reviewer 1 Report

Great descriptive study. Will certainly lead to follow up studies to better understand the microbial dynamics associated with cheese production. Looking forward to seeing those.

Note that the LDA score graphs are not histograms, as they do not depict frequencies. They are bar graphs and this should be corrected in the following versions of the manuscript.

H.

Reviewer 2 Report

This is a very interesting manuscript, it is thematically relevant to the journal "Applied Microbiology". Of course, the task of the reviewer is to detect inaccuracies and deficiencies and ask for their correction or clarification. I have some comments, suggestions and questions for this manuscript, and my opinion says they will help to improve it:

  • I suggest supplementing the information (in the Introduction) on crude extracts of dried Cynara cardunculus L. flower. What does "crude extract" mean? How was this extract obtained from dried flowers? This is extremely important for understanding the microflora found there.
  • Why the S2 cheese samples were only tested in the 1M and 3M period, while the others were tested at 3M and 6M (see figure 1)?
  • The discussion of the results states that the 2018/2019 samples came from manufacturer A, and the 2019/2020 samples came from manufacturer B. In my opinion, this information should be provided in the “Materials and Methods”. Meanwhile, the methodology only provides information on two research periods, which may suggest that the samples came from the same source but from different seasons of cheese production. The clarification of these inaccuracies is very important for the correct interpretation of the results.
  • In my opinion, when discussing the results of the microbial community of raw ewes' milk, it is worth adding information about how the milk is obtained (milking method: manual or mechanical), in what sanitary conditions it takes place (small artisan farms or large industrial farms), There are conditions for storing milk after milking (place, temperature, time, sanitary conditions). This information will help to correctly understand the results of the microbiological quality of raw milk and its safety for cheese making.
  • Discussing the microbial community of cardoon it should be added whether producer A and producer B used extracts in the same source. How distant were they geographically? Could this have been relevant to the observed microbiological differences and similarities in the microbial community of cardoon? The extent to which microflora of flowers reflected in the microflora of extracts from dried flowers?
  • Also, it must be taken into account that samples S1-S3 came from a different manufacturer than samples S4-S6 when interpreting the results of the microbial community of Serra da Estrela cheese. Is the microbiological quality of milk from both producers the same? Are the production conditions and the ripening conditions of the cheeses the same? This is extremely important for the LAB population as well as yeast and mold in cheeses. It should also be taken into account that the ripening stage of the cheeses significantly changes the microflora system in the cheese - usually gradual reduction of LAB population is observed, depending on the genera and species, it takes place to a different extent.

Reviewer 3 Report

Excelent work. It would be very interesting to repeat the work with cheeses proceeding from more producers from different locations of the PDO region. I just have some minor suggestions indicated in the attached file.

Round 2

Reviewer 2 Report

I accept these manuscript versions for publication in "Microorganisms". I have no objection to the corrections introduced by the Authors.